# Meeting the Salinity Requirements of the Bivalve Mollusc *Crassostrea gigas* in the Depuration Process and Posterior Shelf-Life Period to Improve Food Safety and Product Quality

**João A. Silvestre** [1], **Sílvia F. S. Pires** [2], **Vitória Pereira** [2], **Miguel Colaço** [1], **Ana P. L. Costa** [2], **Amadeu M. V. M. Soares** [2], **Domitília Matias** [3], **Filipa Bettencourt** [3], **Sergio Fernández-Boo** [4], **Rui J. M. Rocha** [2,*] and **Andreia C. M. Rodrigues** [2]

1   Department of Biology, University of Aveiro, 3810-193 Aveiro, Portugal; joaosilvestre@ua.pt (J.A.S.); mbfc@ua.pt (M.C.)
2   CESAM (Centre for Marine and Environmental Studies), Department of Biology, University of Aveiro, 3810-193 Aveiro, Portugal; silviapires1@ua.pt (S.F.S.P.); vitoria.pereira@ua.pt (V.P.); anaplcosta@ua.pt (A.P.L.C.); asoares@ua.pt (A.M.V.M.S.); rodrigues.a@ua.pt (A.C.M.R.)
3   IPMA, Instituto Português do Mar e da Atmosfera, 8700-305 Olhão, Portugal; dmatias@ipma.pt (D.M.); fbettencourt@ipma.pt (F.B.)
4   Centro Interdisciplinar de Investigação Marinha e Ambiental (CIIMAR), Universidade do Porto, Avenida General Norton de Matos, S/N, 4450-208 Matosinhos, Portugal; sboo@ciimar.up.pt
*   Correspondence: ruimirandarocha@ua.pt

**Abstract:** Microbiological contamination of bivalve molluscs is one of the major concerns inherent to food safety, thus depuration is frequently needed to assure food safety levels associated with their consumption. Salinity plays an important role in the metabolic activity of bivalves and as such can influence their depuration capacity. This study aimed to evaluate the effect of salinity (25, 30, 35 and 40) on the efficiency of the depuration process, along with the quality and shelf-life of *Crassostrea gigas*. For this, a 24-h depuration was carried out, followed by a storage period at 5 ± 1 °C for six days. Microbiological analyses and biochemical parameters related to oxidative stress response were analysed. *Escherichia coli* load was reduced in only 24 h, disregarding the salinity of the system. After the shelf-life period, the activity of the antioxidant defences at salinities 35 and 40 is higher but is still not sufficient to avoid lipid peroxidation. Over time, there is a decrease in oyster metabolism probably due to being chilled and to the action of exposure to air. In sum, this study suggests salinities between 25 and 30 as preferential for the depuration process of *C. gigas* and subsequent quality during shelf-life.

**Keywords:** bivalves aquaculture; oysters; *Escherichia coli*; salinity; oxidative stress status

## 1. Introduction

Fish and seafood are a vital part of the diet of many people around the world, not only for cultural and traditional reasons but also for health benefits, since they are a good source of protein, vitamins, fatty acids, minerals and other essential micronutrients, hence featuring an excellent nutritional profile [1].

Seafood is a very perishable food that requires proper handling and preservation to ensure its safety, quality and nutritional benefits [2]. Many factors can affect seafood safety, such as the interruption of the cold chain, inadequate equipment, utensils and fishing tackle cleaning, the seafood workers' hygiene, and the lack of control at critical points [3]. Failure to adhere to food-safety best practices during the seafood production chain can trigger microorganism contamination and may result in foodborne diseases [4]. Foodborne infections are a major cause of illness and death worldwide [5,6]. Animal-based foods are widespread worldwide and often considered the key cause of the increase in foodborne infections. Adak et al. [7] found that eating shellfish (a luxury food with relatively low

consumption levels) is associated with very high disease risk. Although the number of cases attributed to shellfish is in the same ranges or levels as beef or eggs, the level of risk is much higher [8].

Responsible aquaculture production of bivalves has a positive environmental impact and significant nutritional benefits, particularly in terms of providing micronutrients. Contrary to intensive fish aquaculture, bivalve aquaculture is an extensive form of aquaculture. These filter-feeders feed on microalgae and other organic matter that occur naturally in the ecosystem, and no extras such as vitamins and antibiotics are added. Additionally, bivalves can boost primary production by increased nutrient recycling [9].

The enlargement of this aquaculture industry has been catalysed by the sustained high prices for bivalves in various regions. Management by farmers is an essential component, whereby they will try to maximise their profits within their aquaculture sites. This is done by growing the bivalves at certain locations where the conditions for growth and survival are maximised [10].

Nevertheless, bivalve molluscan shellfish may cause illness to humans when eaten, since these organisms concentrate contaminants from the water column in which they grow. For microbial contaminants, the threat is enhanced because these shellfish are often eaten raw (e.g., oysters) or relatively lightly cooked (e.g., mussels). Controlling the risk of illness depends in part on obtaining the shellfish from areas in which such contaminants are at fairly low levels [11]. Several factors contribute to the existence of shellfish-associated disease outbreaks, such as problems regarding the monitoring of growing areas and post-harvest contamination of the product (during handling, storing, processing, labelling, and shipping), as well as lack of consumer education and public awareness [1,12].

In the European Union, the Shellfish Water Directive 79/923/EEC regarding the quality of shellfish waters was ratified in 1979. Based on faecal indicator organisms like *E. coli* load and/or *Salmonella* presence/absence, the authorities classify production areas and the treatment required in growing areas during the production cycle and for the end-product. The classification features four classes: Class A ($\leq$230 *E. coli*/100 g), molluscs can be harvested for direct human consumption; Class B (90% of samples must be $\leq$4600 *E. coli*/100 g), molluscs can be sold for human consumption after purification in an approved site, or after re-laying in an approved Class A relaying area, or after an approved heat treatment process; Class C ($\leq$46,000 *E. coli*/100 g), molluscs can be sold for human consumption only after re-laying for at least two months in an approved re-laying area followed, where necessary, by treatment in a depuration centre, or after an approved heat-treatment process; Class D (>46,000 *E. coli*/100 g), molluscs cannot be harvested or sold for human consumption.

Depuration or purification is a process that consists of keeping shellfish in tanks of clean seawater under conditions that maximise the natural filtering activity to expulsion of intestinal contents, enhancing the separation of the expelled impurities from the bivalves, thus avoiding their recontamination [1]. Furthermore, it facilitates the industry's fulfilment of the legal requirements of many countries, which have made depuration of bivalves mandatory under specific circumstances.

There are two kinds of depuration system being utilised: flow-through and recirculating. Flow-through systems are open systems that use natural seawater constantly flowing in the tanks, making it susceptible to fluctuations in the microbial community composition. Recirculating systems are closed systems that require artificial seawater to be constantly cycled through after disinfection. Therefore, commercially used depuration processes are mostly based on keeping the bivalves in closed systems with water recycling via the use of mechanical filters and application of UV radiation, for periods that can vary between 24 and 48 h, regardless of the species, using water temperature values between 10 and 15 °C.

Since one of the goals of this work was to depurate within 24 h, the addition of another treatment element to the commercial standard system was required, namely a skimmer, which allows a foam fractionation. Foam fractionation is an effective water treatment

method that aims to remove proteins, dissolved organic compounds, and other small particles from wastewater. Complying with sustainability guidelines, contemporary aquaculture moves in the direction of a reduced and even zero water exchange in production systems and depuration sites. Thus, effective, easy-to-use floating skimmers that improve water quality are a valuable technological improvement. The technology removes many waste products in recirculating aquaculture systems (RAS) [13].

Oysters of the genus *Crassostrea* are native to inter- and subtidal zones of warm and brackish waters in the western Pacific Ocean but can tolerate and even thrive in a broad extent of salinities and temperatures [14]. Salinity is one of the main environmental factors constraining species survival, biomass, richness and distribution within estuarine systems [15,16]. Generally, *Ostrea* species are distributed in areas with full-strength salinities. Those belonging to the genus *Crassostrea* tend to be more euryhaline, adapting to rapid and dramatic salinity fluctuations from below 10 to almost 35 [17]. Like most marine invertebrates, oysters are osmo-conformers, which implies a lack of ability to osmoregulate the extracellular fluid. Hence, to maintain isosmotic balance, oysters regulate cell volume by accumulating or releasing organic osmolytes (e.g., taurine, betaine) in response to changing salinity. In order to carry out this cell volume and integrity maintenance, oysters have to allocate energy [18–24]. Consequently, this increasing energetic demand during exposure to salinity change may divert resources from other processes, such as defence against pathogens, and even alter the delicate balance between metabolic performance, oxidative stress and energetic fitness that define the organism's acclimation capacity [25].

The main objective of the present work was to assess the influence of salinity on the depuration process and the posterior shelf-life period of the oyster *Crassostrea gigas*. In order to comply with this, the central goal is broken down into two specific sub-objectives: to find out what is the best salinity for the depuration to be effective, and to evaluate the influence of different salinities in the survival, oxidative stress status and quality of the oysters during shelf life.

## 2. Materials and Methods

### 2.1. Sampling Area

The oysters were brought by their producer directly from the production area (zone B), and they were homogeneously sized, partially washed, and experiments were performed not later than 1 h after collection. This production area is located on the western Portuguese Coast (Aveiro), in a relevant region for aquaculture production named the Canal de Mira, part of the Ria de Aveiro. Nowadays, Ria de Aveiro is under several anthropogenic pressures and suffers the influence of a wide range of environmental pollutants, including metals and persistent organic pollutants. Moreover, treated/untreated sewage discharge can also be a major pollutant pathway into this environment [26]. The culture station is located in the Canal de Mira arm of Ria de Aveiro (40.61721 N; −8.74408 W) which receives seawater intrusion but also inland drainage from agriculture fields within the premises of the oyster production farming area. For these reasons, the zone is categorised as a B production zone for *Crassostrea gigas*.

### 2.2. Experimental System

The first step was the assembly and preparation of four closed systems with artificial seawater recirculation to perform the test, in which each worked as an experimental purification module, allowing to test different physical parameters (different salinities).

Each of the modules was equipped with the following components (Figure 1): a 250 L capacity tank; five small tanks of 12 L capacity; a water circulation system (PVC pipes), which allows oxygen levels to be maintained close to saturation values during the depuration process; five PVC taps to control the flow between the PVC pipe and the five small tanks; an EHEIM Universal-pump 1262 recirculation pump, with an output of 3400 L/h, for water impulsion through the system; a chiller (model Hailea HC 300-A (1/4 HP)), for water temperature control; a UV disinfection system (Vecton 600 model); and

finally the differentiating element compared to commercial depuration, a protein skimmer (Eheim, Typ 1103 800) for the removal of organic matter from the water.

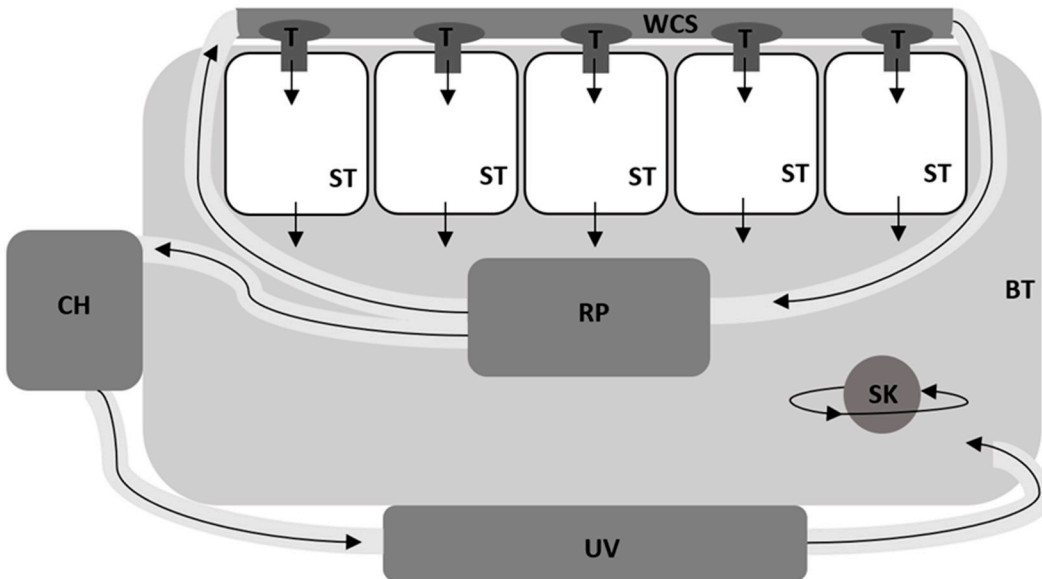

**Figure 1.** Scheme of the depuration module with its different components and circulation movements. Legend: WCS—Water circulation system; BT—Big tank; ST—Small tank; RP—Recirculation pump; CH—Chiller; UV—UV disinfection system; SK—Skimmer; T—Tap.

The water circuit over the entire depuration module (Figure 1) starts with the impulsion of the water through the pump, from the large depuration tank to the chiller, and then on to the UV filter. After UV disinfection by passing through the filter, the water returns to the depuration tank. In parallel with this process, the protein skimmer acts independently, removing proteins and other organic particles dissolved in water. This water is continuously pumped through the circulation system to the small tanks, with the flow being controlled by the taps.

Each system is then considered a 200 L modular system, that allows depuration of up to 30 kg of bivalves within 24 h, thus commercially competing with regular systems at least from a commercial perspective, since regular systems are much bigger (normally 5000 L), consequently having higher depuration costs, therefore being financially unsustainable with small numbers of bivalves. These modular systems enable an operation based on the quantity to be depurated.

### 2.3. Depuration Process

First, a previous assay with temperature variation was performed to define the optimal temperature (unpublished data) to adopt in the main assay of the different salinities, based only on the microbiological analysis of the samples. From this assay, it was decided to adopt 15 °C as the depuration temperature, which is the most common temperature used commercially. Then, the target assay was operated, with salinity variation and the same temperature (15 °C), in order to understand the influence of salinity on this purification process. The chosen salinities were 25, 30, 35, and 40, since they are approximately the values of a regular salinity oscillation in an estuary and possible to reproduce in commercial depurators. It should also be highlighted that the animals were never fed throughout the process. As a control of the depuration assay, the time 0 h, or t(0), which corresponds to individuals without depuration, has been included. For this analysis, a sample of oysters (5 specimens) was taken before the rest were placed in the tanks. This sample was kept at 5 ± 1 °C and then sent in a mesh bag, in an appropriate container, to the IPMA's laboratory (authorised laboratory for analysis of water bacterial quality), in Olhão, for microbiological

analysis of faecal contaminants. In addition to these 5 specimens sent to IPMA, 5 other non-depurated oysters were immediately sampled for posterior biochemical analysis. After 24 h, the system was shut down and all the oysters were removed from each purification module and distributed into mesh bags with different compartments. These bags were moved to a climate chamber and kept at $5 \pm 1$ °C, for 6 days, as in their commercial shelf-life.

### 2.4. Microbiological Analysis

The microbiological analysis was performed in IPMA's laboratory in Olhão, assessing all samples for their faecal contamination using the Donovan-MPN method [27], expressed in MPN/100 g of the tissue of each oyster. Since the oyster had to be alive for this analysis, it is important to emphasise that there was no mortality registered in the transportation of the oysters from the depuration site, in Aveiro, to the laboratory in Olhão.

### 2.5. Experimental Procedure

There were 3 different sampling times: 5 oysters sampled initially (t(0)), which were not depurated; 20 oysters (5 per salinity treatment) after 24 h of depuration; and another 20 oysters after 6 days of shelf-life, i.e., after being kept at $5 \pm 1$ °C for 6 days after depuration. After the asepsis conditions were guaranteed, the oysters were opened, on a regularly sterilised workbench, with the help of a knife separating the shells. Using a scalpel, the edible content of the oyster was removed and distributed in previously marked Falcon tubes, to be immediately frozen in liquid nitrogen, and then kept at $-80$ °C for further biochemical analysis.

As soon as all the oysters had been through this process, the tubes with the edible content were removed from the $-80$ °C fridge and kept on ice. One by one, they were shredded with the help of the scalpel to be later weighed and homogenised. After being weighed, oyster tissue samples were put back into the Falcon tubes and were individually homogenised on ice, using 10 mL of ultra-pure water and the sonicator (pulsed mode of 10% for 30 s, 250 Sonifier, Branson Ultrasonics) [28]. The aim was to study quantitatively the following biochemical biomarkers: Lipids; Sugars; Proteins; Energy available (Ea); Aerobic energy production (Ec); Cellular energy allocation (CEA); Catalase (CAT); Glutathione-S-transferase (GST); Total glutathione (tGSH); and Lipid peroxidation (LPO). Therefore, three aliquots were taken from each sample to analyse lipid, sugar and protein contents, and ETS activity. One aliquot containing 4% butylated hydroxytoluene (BHT) in methanol was used to determine LPO. The remaining homogenate was diluted with 0.2 M K-phosphate buffer, pH 7.4, and centrifuged for 15 min at $10,000 \times g$ (4 °C). The post-mitochondrial supernatant (PMS) was divided into micro tubes and kept at $-80$ °C until further analyses of CAT, GST and tGSH.

All biomarkers determinations were performed spectrophotometrically, in micro-assays set up in 96 well flat-bottom plates, with the Microplate reader MultiSkan Spectrum (Thermo Fisher Scientific, Waltham, MA, USA).

Total lipid content of each organism was determined by adding chloroform, methanol and ultra-pure water in a 2:2:1 proportion, respectively. After centrifugation, the organic phase of each sample was transferred to clean glass tubes, and $H_2SO_4$ was added prior to incubation for 15 min at 200 °C. Absorbance was then measured at 375 nm, and tripalmitin was used as a lipid standard. Carbohydrates quantification was performed by adding 5% phenol and $H_2SO_4$ to the samples, with glucose as a standard, the absorbance read at 492 nm. Bradford's method [29] was used for total protein content quantification using bovine serum albumin as a standard and absorbance measured at 520 nm. Fractions of energy available were converted into energetic equivalent values using the corresponding energy of combustion: 39,500 mJ/g lipid, 17,500 mJ/g glycogen, 24,000 mJ/g protein [30]. Electron transport system (ETS) activity was measured using the INT (Iodonitrotetrazolium) reduction assay, in which ETS was measured as the rate of INT reduction in the presence of the nonionic detergent Triton X-100, with the absorbance read at 490 nm. Cellular oxygen

consumption rate was calculated based on the stoichiometric relationship in which for 2 μmol of formazan formed, 1 μmol of oxygen is consumed. Aerobic energy production (Ec) value was obtained by the conversion to energetic values, using the specific exoenthalpic equivalent for an average lipid, protein and carbohydrate mixture of 480 kJ/mol $O_2$ [31]. Energy available (as the sum of sugars, lipids and proteins) and energy consumption (as ETS activity) were determined by the methods described by De Coen and Janssen [32] with slight modifications for microplate [28]. The final CEA value was then calculated as: CEA = $\frac{Ea}{Ec}$ [33].

According to the Bradford method [30], the protein concentration of PMS was determined using bovine γ-globulin as a standard. Catalase (CAT) activity was determined in PMS by measuring decomposition of the substrate $H_2O_2$ at 240 nm [34]. Glutathione-S-transferase (GST) activity was determined in PMS following the conjugation of GSH with 1-chloro-2,4-dinitrobenzene (CDNB) at 340 nm [35]. Total glutathione (TG) content was determined with PMS fraction at 412 nm using a recycling reaction of reduced glutathione (GSH) with 5,5′-dithiobis-(2-nitrobenzoic acid) (DTNB) in the presence of glutathione reductase (GR) excess [36,37]. tGSH content was calculated as the rate of $TNB^{2-}$ formation with an extinction coefficient of DTNB chromophore formed, $\varepsilon = 14.1 \times 103\ M^{-1}cm^{-1}$ [35,37]. Endogenous lipid peroxidation (LPO) was determined by measuring thio-barbituric acid-reactive substances (TBARS) at 535 nm [38].

*2.6. Statistical Analysis*

For each data set, outliers (ROUT (Q = 1%)) and normality (Shapiro-Wilk's W test) were tested, and transformations (Y = sqrt(Y) or Y = ln(Y)), if necessary, were applied. In order to understand the variances of the data between the samples of t(0) and t(24), Student's *t*-test was performed. To assess whether salinity and/or time of shelf life caused a different response for the biomarker under analysis, two-way ANOVA was run with two different multiple comparison tests—one assessing the difference between salinities and the other different times. Both multiple comparisons were Ŝídák tests with a 95% confidence interval. A full description of the statistical analysis values can be found in the Supplementary Materials, Tables S1 and S2. The statistical analysis was performed using Graphpad Prism software version 9.0.

## 3. Results

No mortality was recorded throughout the assay: all specimens were alive at the time of sampling, at 0 h, 24 h and 6 days of shelf-life.

A full description of the values from the statistical analysis is expressed in Supplementary Tables S1 and S2.

*3.1. Microbiological Results*

There was a clear decrease in the contamination by *E. coli* in every salinity, as shown in Table 1. The number of E. coli in MPN/100 g decreased from values that would not allow the immediate consumption (zone B) of the oysters to values below the detection limit of the method of quantification in every salinity after 24 h of depuration.

**Table 1.** Microbiological analysis (MPN *E. coli*/100 g) of the *Crassostrea gigas* oysters at arrival in the laboratory (T 0 h) and after 24 h depuration at the tested salinities.

| Sampling Times | T 0 h | | T 24 h | | | | | | | |
|---|---|---|---|---|---|---|---|---|---|---|
| Experimental salinities | - | - | 25 | 25 | 30 | 30 | 35 | 35 | 40 | 40 |
| *E. coli* (MPN/100 g) | 490 | 230 | 20 | <18 | 20 | 20 | <18 | 20 | <18 | <18 |

### 3.2. Oxidative Stress Related Biomarkers

#### 3.2.1. 24-h Depuration Period

Regarding the energetic fitness of the oysters, a marginally significant ($p = 0.076$) and a significant ($p = 0.014$) increase in the sugar values of animals was observed depurated (t(24 h)) at salinity 35 and 40, respectively (Figure 2B), when compared with non-depurated animals. Lipids (Figure 2A) and proteins (Figure 2C) did not significantly differ between the non-depurated oysters and those depurated for 24 h at different salinities. As expected, since the available energy is the sum of the three previous parameters, only a marginally significant ($p = 0.052$) difference between oysters from t(0) and depurated for 24 h in water of salinity 40 was observed (Figure 2D). There was a marginally significant ($p = 0.056$) decrease regarding aerobic energy production between the oysters at t(0) and the depurated oysters for 24 h in salinity 25 (Figure 2E). Concerning the cellular energy allocation, which is the ratio between available and consumed energy, all the values of the oysters depurated for 24 h at the different salinities were significantly higher than the t(0) ($p = 0.036$, $p = 0.013$, $p = 0.009$, $p = 0.039$, to comparisons between t0 and salinities 25, 30, 35 and 40, respectively, Figure 2F).

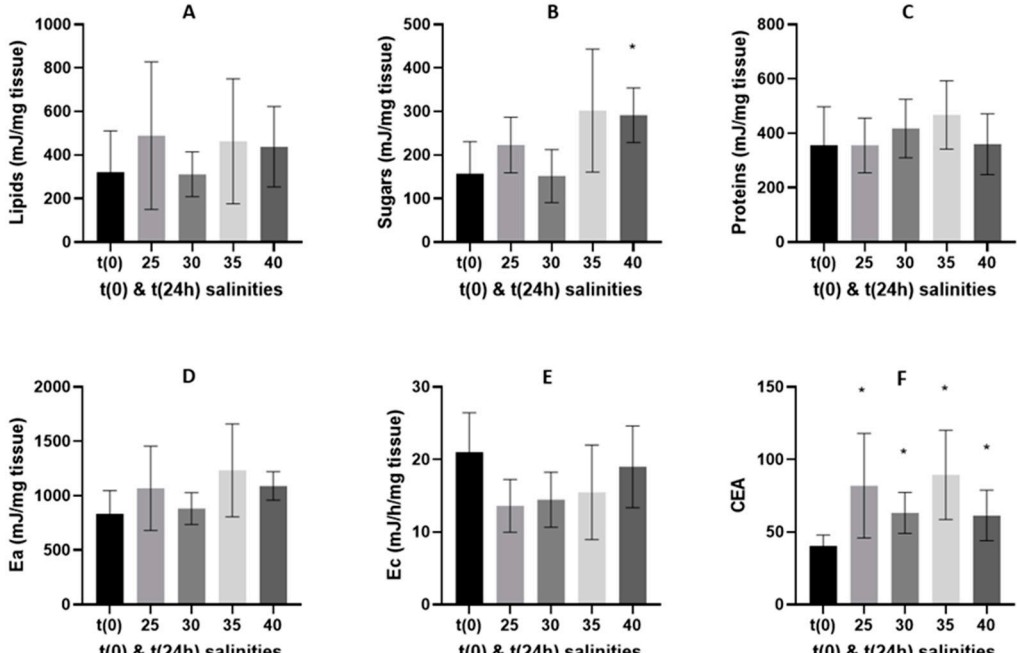

**Figure 2.** Cellular energy allocation parameters of non-depurated and depurated *Crassostrea gigas* (t0) for 24 h at the 4 tested salinities: (**A**)—Lipids; (**B**)—Sugars; (**C**)—Proteins; (**D**)—Energy available; (**E**)—Consumed energy; (**F**)—Cellular energy allocation; * *t*-test significant difference, $p < 0.05$. Data is presented as mean $\pm$ SD.

Observing the biomarkers related to antioxidant defences (Figure 3), the values for catalase activity were marginally significant ($p = 0.061$) and significantly ($p = 0.008$) higher in the oysters depurated at salinities 35 and 40 respectively (Figure 3A), when compared with non-depurated oysters. For tGSH, in salinity 35 a significant increased ($p = 0.032$) was observed when compared to those of t(0), and a marginally significant ($p = 0.056$) difference was observed compared with salinity 40 (Figure 3B). Even though there were no significant differences between the values of GST activity of the depurated oysters from every salinity and the values of the non-depurated oysters, t(0), marginally significant ($p = 0.056$) higher GST activity with increasing salinity can be observed (35 and 40, Figure 3C). As for LPO, apart from a clear tendency to increase with increasing salinity, there is a significant ($p = 0.008$) decrease after 24 h of depuration comparing the values of the non-depurated samples, t(0), with those depurated at 25 salinity (Figure 3D).

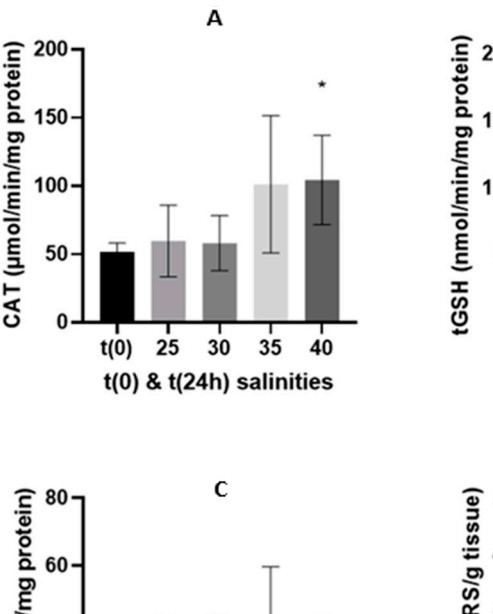
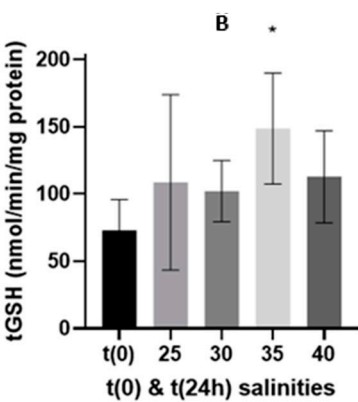
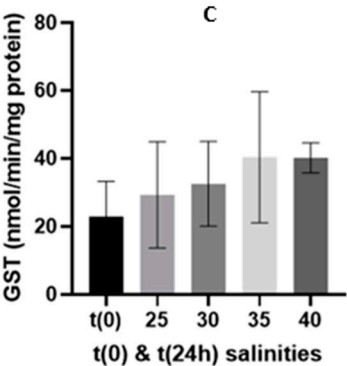
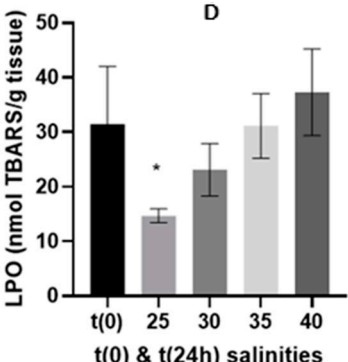

**Figure 3.** Oxidative stress-related biomarkers of non-depurated and depurated *Crassostrea gigas* (t(0)) for 24 h at the 4 tested salinities: (**A**)—Catalase activity; (**B**)—Total Glutathione levels; (**C**)—Glutathione-S-transferase activity; (**D**)—Lipid peroxidation levels; * *t*-test significant difference, $p < 0.05$. Data is presented as mean ± SD.

3.2.2. Six Days of Shelf-Life

There were no significant differences caused by time or different salinities concerning the energetic fitness of oysters from 24 h depuration to six days of shelf-life (Figure 4). However, a marginally significant ($p = 0.064$) interaction effect on the sugar values was translated into an inversion of sugar content from 24 h of depuration to six days of shelf-life, with the change of the highest sugar values from the lowest to the highest salinities (Figure 4B). The same happened to the aerobic energy production pattern (Figure 4E).

In terms of oxidative stress biomarkers, namely the catalase activity, there is a clear inversion of the tendency of values between the 24 h of depuration and the six days of shelf-life, so the interaction factor is significant ($p = 0.004$). Despite this significant variance, it was not possible to discriminate significant differences between groups with the multiple comparisons test (Figure 5A). There was a significant ($p = 0.001$) decrease over time in the tGSH values. A significant difference occurred in salinity 35, between the two different times, with a clear decrease in tGSH activity from 24 h of depuration to six days shelf life (Figure 5A).

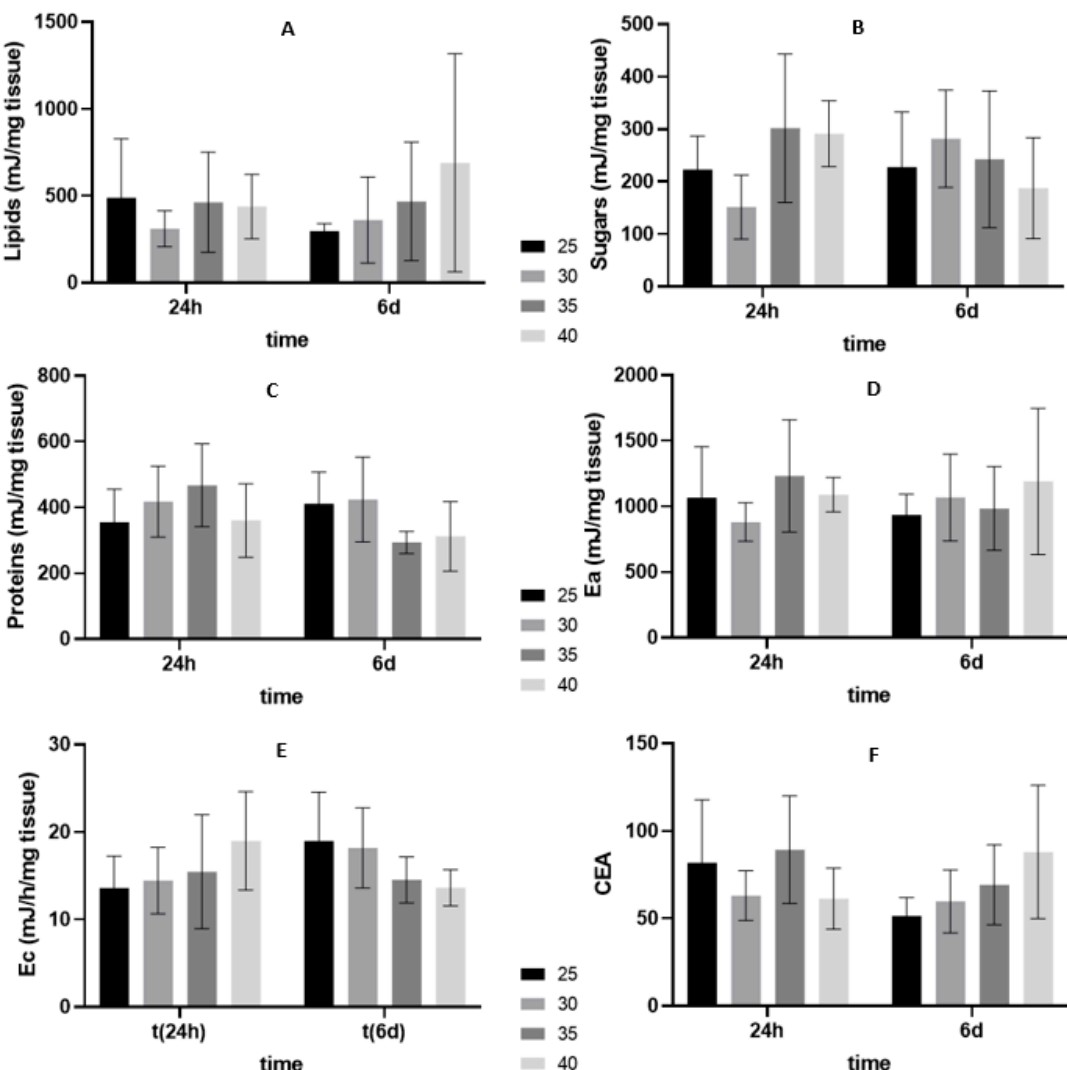

**Figure 4.** Cellular energy allocation parameters of *Crassostrea gigas* oysters immediately after the 24 h depuration and after 6 days shelf-life at the 4 tested salinities: (**A**)—Lipids; (**B**)—Sugars; (**C**)—Proteins; (**D**)—Energy available; (**E**)—Consumed energy; (**F**)—Cellular energy allocation. Data is presented as mean ± SD.

GST activity values were significantly ($p$ = 0.016) changed due to the interaction of time and salinities (Figure 5C). Within 24 h, the GST values do not present differences among the different salinities, but the same does not happen for the six days of shelf-life. Here, the significant difference between the values recorded at 30 salinity and the two highest salinity values (35 and 40) stands out. It is also possible to denote that the greatest difference between the 24 h of depuration and the six days of shelf-life in terms of GST activity was also at 30 salinity, with a significant difference between the same treatment in the two different times, also represented in Figure 5C.

Finally, analysing the LPO levels, a significant effect of salinity ($p$ < 0.001) and its interaction ($p$ < 0.001) with time was observed in *C. gigas*. It is possible to verify that there is an approximation of the LPO values between salinities at six days of shelf-life, compared to 24 h, where the LPO values increased according to the increase in salinity (Figure 5D). There are also significant differences between the LPO values of oysters depurated at 25 and 40 salinity between the two different times.

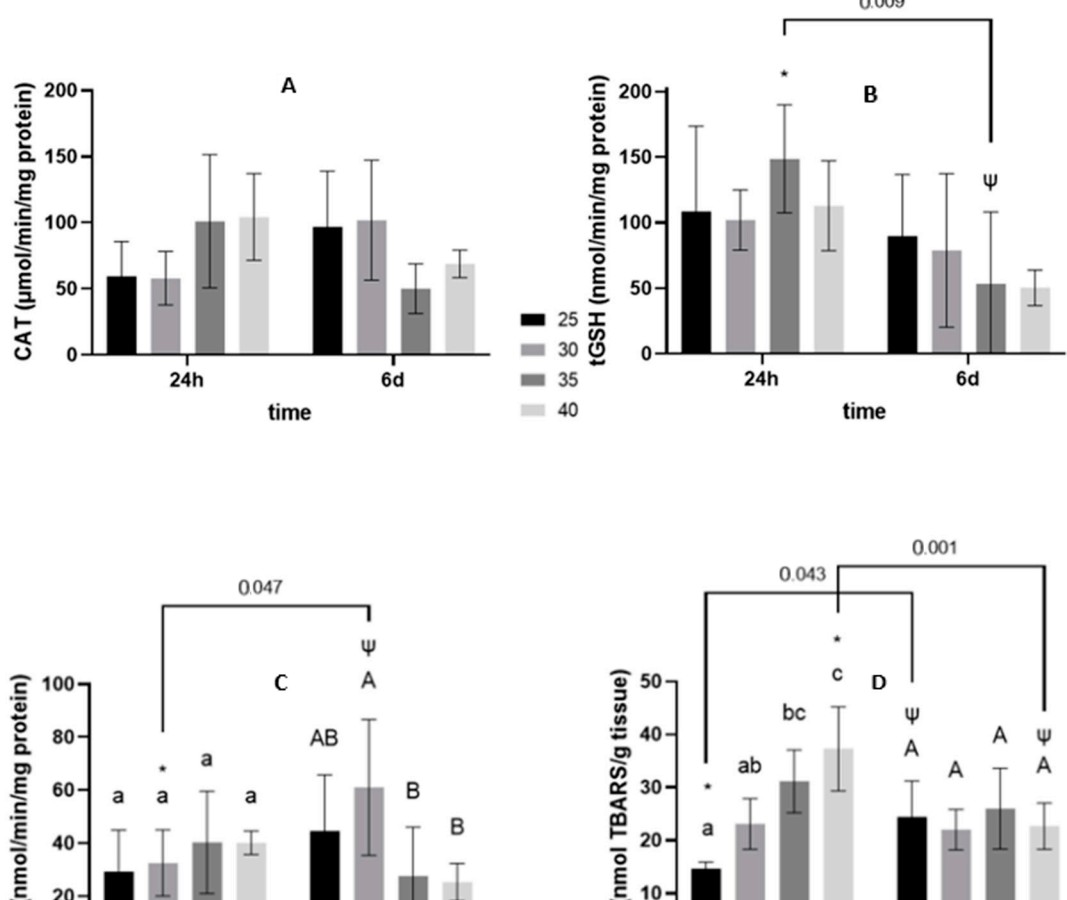

**Figure 5.** Oxidative stress-related biomarkers of *Crassostrea gigas* oysters immediately after the 24 h depuration and 6 days of shelf-life at the 4 tested salinities: (A—Catalase activity; (**B**)—Total Glutathione levels; (**C**)—Glutathione-S-transferase activity; (**D**)—Lipid peroxidation levels; Different letters—Significant difference within different salinities; Different symbols—Significant difference over time. Data is presented as mean ± SD.

## 4. Discussion

### 4.1. Microbiological Analysis

The values of the microbiological analysis obtained in this study showed that the oysters' depuration was effective after 24 h for all tested salinities. The oysters that were analysed before depuration contained *Escherichia coli* levels that would make direct human consumption impossible, but after 24 h under depuration with the said conditions and materials, the oysters became suitable for commercialisation. The effectiveness of the depuration process depends on the design of the set-up, species [39], physiological condition [40], initial concentration of bacteria [41], water temperature and salinity in the tanks [42].

The UV filter, responsible for the elimination of pathogens, was efficient within 24 h, with which the action of the skimmer can be associated. The use of UV radiation has some disadvantages, namely the requirement of good physical filtration to remove suspended material [43,44]. The use of skimmer filtration seems to have been sufficient to remove suspended material and, therefore, to maintain the effectiveness of the UV filtration process, which may explain the significant decrease of *E. coli* values in only 24 h of depuration.

These results are not surprising given that this specific oyster species is considerably euryhaline, being tolerant and able to adapt to a wide range of salinities [15]. Hence, if this

study was solely based on the microbiological data, it would not be possible to determine the optimum salinity for depuration, since the depuration has been very effective in all tested salinities with very similar values (20 MPN/100 g in some cases and even below 18 MPN/100 g, which is the minimum value detectable by the method). However, this study also addresses the biomarkers' analysis, to understand precisely if there were sub-lethal alterations at the cellular level of the organisms to achieve this optimal salinity for the depuration.

### 4.2. Energy Budget and Oxidative Stress-Related Biomarkers

The increase in sugar values after 24 h depuration, without food added, suggests that the highest salinities (35 and 40) stimulated gluconeogenesis—the metabolic process by which organisms produce sugars (namely glucose) for catabolic reactions from non-carbohydrate precursors. Since, in general, sugars, or carbohydrates, form the main energy source for the body, being the most efficient at producing ATP, they are the first to be broken down in metabolism [45]. Therefore, to respond to stressful situations like high salinity, such as 35 and 40, by increasing antioxidant defences, and since they had no external source of food, oysters had to expend energy through glucose, thus performing gluconeogenesis. At six days of shelf time, and probably due to ice lowering the metabolism, a change in the use of sugars was no longer observed, as the animal no longer consumes so much energy.

After sugars, the body starts to break down the lipids to produce ATP and finally the proteins—generally less efficient at generating energy—but only if the other two are depleted. There were no significant differences between salinities and shelf life regarding these two components, showing that the stress, even at these salinities, was not extreme. Consistent with this outcome, Freitas et al. [46] showed in their study with the species *Ruditapes philippinarum* exposed to different salinities (14, 21, 28, 35 and 42) for 96 h, after a two-week acclimation period at salinity 28 (salinity of the sampling site), that along the salinity gradient this species tended to maintain its protein content, with the highest content occurring at the highest salinity (42) compared to the control salinity of 28. Protein synthesis may occur in response to high salinity to reduce the concentration of amino acid in the cell so that the cell remains isotonic with the surrounding seawater. Intracellular concentration of free amino acids is an important parameter for controlling cell volume in response to changes in salinity in osmo-conformers [24,47,48]. Such compensatory mechanisms allow acclimation to salinity, but they can affect energy status [49]. Nonetheless, the animals were able to roughly maintain energy balance and performance [50].

Since there were no differences in the available energy, whose tendency follows the sugar pattern, it is curious to observe the parameter of aerobic energy production, which corresponds to what was previously reported. In the salinity of 40 and t(0), the oysters needed to increase aerobic respiration to produce ATPs in order to fight stressful conditions. Animals exposed to salinity stress must increase their energy expenditure to successfully acclimate to the stressor and ensure cellular protection [51]. Velez et al. [52] have demonstrated that the clam *R. decussatus* exposed to salinity 14 presented increased metabolic activity (ETS). Similarly, the oyster *Crassostrea angulata* presented increased ETS activity at low salinities (10 and 20), and these results were attributed to higher metabolic rates associated with hypo-osmotic stress [53]. The high energy value consumed in t(0) is justified due to the fact that individuals are exposed to air, the worst possible condition for an aquatic animal. In this 24 h-period, when oysters are put back into water after transport to the depuration site, there is a calming of antioxidant responses. After six days of shelf time, the oysters that had lower levels of stress at the beginning maintain slightly higher levels of aerobic respiration, while those of higher salinities show a lower level of energetic consumption, which may reflect long term effects of the energy costs to deal with higher surrounding salinity [22,54,55]. Likewise, Kim et al. [55] observed increased oxygen consumption in *R. philippinarum* clams exposed to salinity stress [23], indicating metabolic adjustment to hypoosmotic stress. This protective strategy against osmotic stress

is in favour of the conservation of energy reserves by reducing respiration, also supported by the fact that they were on ice during these days, which also decreased metabolism.

The CEA shows significantly different values when comparing non-depurated oysters with all other salinities due to the fact that there was a high energy consumption for oysters that were not replaced in water and were always exposed to air. Erk et al. [56] exposed individuals of the species *Neomysis integer* to multiple salinities (5, 17 and 26) and a concentration of cadmium for seven days, and it was noticed that CEA was affected by salinity and the duration of the experiment, with all salinities decreasing over time, and this could be explained by increased energy consumption.

It is known that marine bivalves exposed to stressful conditions increase reactive oxygen species (ROS) production, with a consequent possible increase in lipid, DNA and protein damage if an increase in the antioxidant and detoxification mechanisms is not accomplished [57,58]. Regarding antioxidant defences, it was clear that at the highest salinities there was greater production of reactive oxygen species (ROS) and consequently higher activity of catalase, glutathione-S-transferase and total glutathione. These results are consistent with other stress factors, such as high temperature, as has been shown in previous studies of oxidative stress in marine environments [59–61]. In line with these findings, in Khessiba et al. [62], using the Mediterranean mussel species (*Mytilus galloprovincialis*) for four days, it was also observed that higher salinities increased CAT activity. In addition, other authors [63–65] also observed a GST activity increase in organisms exposed to diverse contaminants, namely PAH, PCBs, furans, phenobarbital compounds and others. Moreover, changes in total glutathione can be related to the role of several hormones, such as catechols and estrogens [66]. Even though there was higher activity of the antioxidant defences, it was not enough to deal with the forming ROS, thus occurring oxidative damage at the lipidic level at both 35 and 40 salinities. A similar response was observed by Velez et al. [52] who showed that although *R. philippinarum* presented higher activity of antioxidant enzymes at low pH conditions, LPO increased, suggesting the insufficient production of these enzymes to eliminate the excess of ROS produced. In Zanette et al. [67], where salinities of 9, 15, 25 and 35 were used to assess the vulnerability of *C. gigas* to oil exposure, it was also observed that salinity changes highly influenced lipid peroxidation. After six days of shelf-life, LPO levels are explained by the balance between air exposition and low metabolism due to permanence on ice, leading to homogeneity within animals previously depurated at different salinities.

## 5. Conclusions

In this study, unlike the commercial depuration method that normally takes up to 48 h, it was shown that depuration is effective in half of this time, 24 h, thus leading to a clear cost reduction with lower energy expenditure, thus being more sustainable, and to a faster intermediate process between producer and consumer.

As expected, the biochemical biomarkers at the cell level proved to be more sensitive than the microbiological analysis. From the microbiological data, it could be assumed that there was no difference between salinities, but by analysing the data set of these biomarkers, it is possible to indicate the salinities of 25 and 30 as the best values for *Crassostrea gigas*.

Undoubtedly, the large diversity of marine invertebrates and the great variability in osmoregulatory strategies, life histories, evolutionary backgrounds, and tissue functions are responsible for the enormous variability of energy–redox responses upon changes in environmental salinity. Most importantly, understanding such subcellular processes may help us elucidate different evolutionary adaptations to different marine environments and predict the role of ROS in promoting or preventing essential physiological responses leading to stress acclimation. In the future, it is important to carry out studies to determine abiotic factors (e.g., salinity and temperature) for other bivalve species, not only to increase the quality and the shelf-life but also to reduce the costs associated with depuration while ensuring food safety aspects. Little is known about seafood workers' knowledge, beliefs, and habits regarding seafood handling and safety. Therefore, several studies that

assess seafood handling practices by seafood workers are needed, to ensure or improve seafood safety.

**Supplementary Materials:** The following are available online at https://www.mdpi.com/article/10.3390/w13081126/s1, Table S1. Parametric *t*-test or non-parametric Mann-Whitney U values for the measured biochemical biomarkers of *C. gigas* of the t(0) and each salinity of the t(24 h); Table S2. Two-way ANOVA parameters showing total variation (%), F and *p* values for the biochemical biomarkers measured in *C. gigas* after 24 h depuration and 6 days of shelf-life, with time and salinity as factors.

**Author Contributions:** Conceptualisation, R.J.M.R., A.C.M.R., D.M.; methodology and investigation, J.A.S., M.C., A.P.L.C., A.C.M.R., V.P., S.F.S.P., F.B.; formal analysis, J.A.S., A.C.M.R.; writing—original draft preparation, J.A.S.; writing—review and editing, A.C.M.R., R.J.M.R., S.F.-B.; supervision, R.J.M.R., A.C.M.R.; project administration, R.J.M.R., S.F.-B.; funding acquisition, R.J.M.R., S.F.-B., A.M.V.M.S. All authors have read and agreed to the published version of the manuscript.

**Funding:** Thanks are due for the financial support to CESAM (UIDB/50017/2020 + UIDP/50017/2020), to FCT/MEC through national funds. This work is part of the project Biodepura (MAR-02.01.01-FEAMP-0018), supported by Portugal and the European Union through FEAMP, MAR2020 in the framework of Portugal 2020. This work was also supported by the project POCI-01-0145-FEDER-030232, co-financed by COMPETE 2020, Portugal 2020 and the European Union through the ERDF. This research was supported by national funds "through FCT—Foundation for Science and Technology" within the scope of UIDB/0443/2020 and UIDP/04423/2020.

**Institutional Review Board Statement:** Not applicable.

**Informed Consent Statement:** Not applicable.

**Data Availability Statement:** The data presented in this study is available in the current manuscript, raw data is available on request.

**Conflicts of Interest:** The authors declare no conflict of interest.

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
