# Peer review of "Meeting the Salinity Requirements of the Bivalve Mollusc Crassostrea gigas in the Depuration Process and Posterior Shelf-Life Period to Improve Food Safety and Product Quality"

_water, doi:10.3390/w13081126_

Round 1

Reviewer 1 Report

The paper aimed to assess the influence of salinity in the depuration process and the posterior shelf-life period (after 6 days) of the oyster Crassostrea gigas. In order to comply with this, the central goal is broken down into two specific sub-objectives:  find out what is the best salinity for the depuration to be effective; evaluate the influence  of different salinities in the survival, oxidative stress status and quality of the oysters during shelf life.

The work is good for publication and innovative regarding the targeted species and the location.

The work is well written and clear. I suggest checking in the text that the species name is written in italics.

Line 270 "only a marginally significant" the author use often this sentence. I don't understand "marginally" if is not significant.

Moreover, is better to added the p-valuein the text, this can help identify when there is a significant difference.

Author Response

The paper aimed to assess the influence of salinity in the depuration process and the posterior shelf-life period (after 6 days) of the oyster Crassostrea gigas. In order to comply with this, the central goal is broken down into two specific sub-objectives:  find out what is the best salinity for the depuration to be effective; evaluate the influence  of different salinities in the survival, oxidative stress status and quality of the oysters during shelf life.

The work is good for publication and innovative regarding the targeted species and the location.

The work is well written and clear. I suggest checking in the text that the species name is written in italics.

Authors: Species names were checked. Thanks.

Line 270 "only a marginally significant" the author use often this sentence. I don't understand "marginally" if is not significant.

Authors: The expression marginally significant is used to refer to p values between 0.1 and 0.05. It is a question that has been largely debated within the scientific community, and we agree that when it regards biological meaning of the data, p values below low values such as 0.1 may be taken into consideration.

Moreover, is better to added the p-value in the text, this can help identify when there is a significant difference.

Authors: p values were now added. Thanks.

Reviewer 2 Report

Review

Paper title: Meeting the Salinity Requirements of the Bivalve Mollusc Crassostrea Gigas in the Depuration Process and Posterior Shelf-Life Period to Improve Food Safety and Product Quality.

Bivalve molluscs in general and Crassostrea gigas in particular are important products in human diets as they are rich in protein and essential fatty acids. Aquaculture production of bivalve molluscs has been shown to increase over the last decades. Being suspension feeders, bivalves are able to accumulate harmful contaminants from water. Thus, depuration is an important process that is used to reduce the potential toxicity of cultured molluscs. The authors studied the effects of salinity on the depuration process and the posterior shelf-life period of the oyster Crassostrea gigas. They suggested that the salinity levels of 25 and 30 are optimal for this species in terms of the survival rate, oxidative stress status, and quality of the oysters during shelf life.

All these reasons explain the relevance of the paper by J. A. Silvestre and co-authors. submitted to "Water".

General scores.

The data presented by the authors are original and significant. All conclusions are justified and supported by the results. The study is correctly designed and technically sounds. In general, the statistical analyses are performed with good technical standards. We authors conducted careful work which will attract the attention of a wide range of specialists focused on the biology of mussels, especially biological monitoring of water bodies and food technologists.

I recommend this paper for publication after minor revisions.

Specific comments.

L 38. Change “algae” to “microalgae”

L 51. Change “shellfish associated diseases” to “shellfish-associated disease”

L 64. Change “2 month” to “2 months”

L 74. Change “depuration system” to “depuration systems”

L 83. Change “in 24h” to “within 24 h”

L 96. Change “full strength” to “full-strength”

L 102. Change “[11,12,13,14,15,16,17]” to “[11–17]”

L 107. Change “salinity in” to “salinity on”

L 110. Change “; evaluate” to “and to evaluate”

L 117. Change “no > 1h after collection” to “not later than within 1h after collection”

L 125. Change “For all this” to “For these reasons”

L 125. Change “Each of the modules” to “Each module”

L 131. Change “Figure 1.” to “Figure 1”

L 140. Change “Figure 1.” to “Figure 1”

L 147. Change “Each one of these systems” to “Each system”

L 148. Change “in 24h” to “within 24 h”

L 174-175. Change “during 6 days” to “for 6 days”

L 178-179. Change “Donovan-MPN method according to [20]” to “Donovan-MPN method [20]”

L 205. Change “kept in” to “kept at”

L 228. Change “by [24]” to “by De Coen and Janssen [24]”

L 245. Change “t-test” to “Student’s  t-test”

L 245. Change “95% confidence” to “a 95% confidence”

L 258. " E. coli" should be italicized.

L 261. Change “arrival at” to “arrival in”

L 271. Change “at 40” to “in water of salinity 40”

L 278. Change “C. gigas” to “Crassostrea gigas”

L 281. Change “Figure 3.” to “Figure 3”

L 284. Change “significant increased” to “significant increase”

L 294. Change “C. gigas” to “Crassostrea gigas”

L 297. Change “3.2.2.6. days of Shelf-Life” to “3.2.2. Six days of Shelf-Life”

L 299. Change “Figure 4.” to “Figure 4”

L 300. Change “interaction” to “interaction effect”

L 305. Change “C. gigas” to “Crassostrea gigas”

L 331. Change “C. gigas” to “Crassostrea gigas”

L 345. Change “in 24h” to “within 24 h”

L 376. Change “[39] showed” to “Freitas et al. [39] showed”

L 386. Change “[43] roughly” to “roughly [43]”

L 392. Change “[45] demonstrated” to “Velez et al. [45] have demonstrated”

L 395. Change “results” to “these results”

L 397. Change “In this 24h” to “In this 24h-period”

L 400. Change “maintaining” to “maintain”

L 403. Change “[48] observed” to “Kim et al. [48]  observed”

L 406. Change “affected” to “supported”

L 410-411. Change “A study by [49] has exposed” to “Erk et al. [49] exposed”

L 416. Change “consequent possible increase” to “a consequent possible increase”

L 418. Change “acomplished” to “accomplished”

L 421. Change “temperature, in previous” to “temperature as has been shown in previous”

L 423. Change “by [55]” to “Khessiba et al. [55] ”

L 424. Change “In addition, [56,57,58]” to “In addition, other authors [56,57,58]”

L 428. Change “a higher activity” to “higher activity”

L 430. Change “35 and 40” to “35 and 40 salinities”

L 430. Change “by [45] that showed” to “by Velez et al [45] who showed”

L 443. Change “cell level” to “the cell level”.

The authors should cite their supplementary tables in the “Results” section.

The authors should mention in the figure captions that vertical bars show 95% confidence intervals.

References should be formatted according to Instructions for authors.

Author Response

Paper title: Meeting the Salinity Requirements of the Bivalve Mollusc Crassostrea Gigas in the Depuration Process and Posterior Shelf-Life Period to Improve Food Safety and Product Quality.

Bivalve molluscs in general and Crassostrea gigas in particular are important products in human diets as they are rich in protein and essential fatty acids. Aquaculture production of bivalve molluscs has been shown to increase over the last decades. Being suspension feeders, bivalves are able to accumulate harmful contaminants from water. Thus, depuration is an important process that is used to reduce the potential toxicity of cultured molluscs. The authors studied the effects of salinity on the depuration process and the posterior shelf-life period of the oyster Crassostrea gigas. They suggested that the salinity levels of 25 and 30 are optimal for this species in terms of the survival rate, oxidative stress status, and quality of the oysters during shelf life.

All these reasons explain the relevance of the paper by J. A. Silvestre and co-authors. submitted to "Water".

General scores.

The data presented by the authors are original and significant. All conclusions are justified and supported by the results. The study is correctly designed and technically sounds. In general, the statistical analyses are performed with good technical standards. We authors conducted careful work which will attract the attention of a wide range of specialists focused on the biology of mussels, especially biological monitoring of water bodies and food technologists.

I recommend this paper for publication after minor revisions.

Specific comments.

L 38. Change “algae” to “microalgae”

L 51. Change “shellfish associated diseases” to “shellfish-associated disease”

L 64. Change “2 month” to “2 months”

L 74. Change “depuration system” to “depuration systems”

L 83. Change “in 24h” to “within 24 h”

L 96. Change “full strength” to “full-strength”

L 102. Change “[11,12,13,14,15,16,17]” to “[11–17]”

L 107. Change “salinity in” to “salinity on”

L 110. Change “; evaluate” to “and to evaluate”

L 117. Change “no > 1h after collection” to “not later than within 1h after collection”

L 125. Change “For all this” to “For these reasons”

L 125. Change “Each of the modules” to “Each module”

L 131. Change “Figure 1.” to “Figure 1”

L 140. Change “Figure 1.” to “Figure 1”

L 147. Change “Each one of these systems” to “Each system”

L 148. Change “in 24h” to “within 24 h”

L 174-175. Change “during 6 days” to “for 6 days”

L 178-179. Change “Donovan-MPN method according to [20]” to “Donovan-MPN method [20]”

L 205. Change “kept in” to “kept at”

L 228. Change “by [24]” to “by De Coen and Janssen [24]”

L 245. Change “t-test” to “Student’s  t-test”

L 245. Change “95% confidence” to “a 95% confidence”

L 258. " E. coli" should be italicized.

L 261. Change “arrival at” to “arrival in”

L 271. Change “at 40” to “in water of salinity 40”

L 278. Change “C. gigas” to “Crassostrea gigas”

L 281. Change “Figure 3.” to “Figure 3”

L 284. Change “significant increased” to “significant increase”

L 294. Change “C. gigas” to “Crassostrea gigas”

L 297. Change “3.2.2.6. days of Shelf-Life” to “3.2.2. Six days of Shelf-Life”

L 299. Change “Figure 4.” to “Figure 4”

L 300. Change “interaction” to “interaction effect”

L 305. Change “C. gigas” to “Crassostrea gigas”

L 331. Change “C. gigas” to “Crassostrea gigas”

L 345. Change “in 24h” to “within 24 h”

L 376. Change “[39] showed” to “Freitas et al. [39] showed”

L 386. Change “[43] roughly” to “roughly [43]”

L 392. Change “[45] demonstrated” to “Velez et al. [45] have demonstrated”

L 395. Change “results” to “these results”

L 397. Change “In this 24h” to “In this 24h-period”

L 400. Change “maintaining” to “maintain”

L 403. Change “[48] observed” to “Kim et al. [48]  observed”

L 406. Change “affected” to “supported”

L 410-411. Change “A study by [49] has exposed” to “Erk et al. [49] exposed”

L 416. Change “consequent possible increase” to “a consequent possible increase”

L 418. Change “acomplished” to “accomplished”

L 421. Change “temperature, in previous” to “temperature as has been shown in previous”

L 423. Change “by [55]” to “Khessiba et al. [55] ”

L 424. Change “In addition, [56,57,58]” to “In addition, other authors [56,57,58]”

L 428. Change “a higher activity” to “higher activity”

L 430. Change “35 and 40” to “35 and 40 salinities”

L 430. Change “by [45] that showed” to “by Velez et al [45] who showed”

L 443. Change “cell level” to “the cell level”.

Authors: Altered as suggested. Thanks.

The authors should cite their supplementary tables in the “Results” section.

Authors: The content of supplementary tables was now reinforced at the beginning of the results section: “A full description of the values from the statistical analysis is expressed on the supplementary tables SI and SII.”

The authors should mention in the figure captions that vertical bars show 95% confidence intervals.

Authors: The information about data presentation was now added to the figure captations: “Data is presented as mean ± SD.”

References should be formatted according to Instructions for authors.

Authors: References were checked and formatted accordingly. Thanks.

Reviewer 3 Report

The paper is very interesting and I have read it with interest.

I have only a couple of suggestions, that may be taken to improve the paper.

First, the paper should better discuss potential issues related to food contamination, and risk to which consumers would be exposed. Also, the authors may want to deepen on the subjectivity of food safety (i.e. the importance of communicate the safety of foods, especially when seafood is considered). On these aspects, the authors may refer to recent works such as 

Santeramo, F. G., & Lamonaca, E. (2020). Objective risk and subjective risk: The role of information in food supply chains. Food Research International

Zanin, L. M., da Cunha, D. T., Stedefeldt, E., & Capriles, V. D. (2015). Seafood safety: knowledge, attitudes, self-reported practices and risk perceptions of seafood workers. Food Research International67, 19-24.

Second, the authors may want to broaden their discussion with implications for the whole food industr, rather than referring only to a specific portion of the sector. This addition would be feasible by adding more references in the last section, so as to compare the implications of the study with recent studies.  

Author Response

The paper is very interesting and I have read it with interest.

I have only a couple of suggestions, that may be taken to improve the paper.

First, the paper should better discuss potential issues related to food contamination, and risk to which consumers would be exposed. Also, the authors may want to deepen on the subjectivity of food safety (i.e. the importance of communicate the safety of foods, especially when seafood is considered). On these aspects, the authors may refer to recent works such as 

Santeramo, F. G., & Lamonaca, E. (2020). Objective risk and subjective risk: The role of information in food supply chains. Food Research International

Zanin, L. M., da Cunha, D. T., Stedefeldt, E., & Capriles, V. D. (2015). Seafood safety: knowledge, attitudes, self-reported practices and risk perceptions of seafood workers. Food Research International67, 19-24.

Second, the authors may want to broaden their discussion with implications for the whole food industr, rather than referring only to a specific portion of the sector. This addition would be feasible by adding more references in the last section, so as to compare the implications of the study with recent studies.  

Authors: We appreciated the reviewer suggestions. Based on the suggested works we found extra references and completed the Introduction and conclusions sections, respectively:

Page1-2 LL 38-48: “Seafood is a very perishable food that requires proper handling and preservation to ensure its safety, quality and nutritional benefits [2]. Many factors can affect seafood safety, such as the interruption of the cold chain, inadequate equipment, utensils and fishing tackle cleaning, the seafood workers' hygiene, and the lack of control at critical points [3]. Failure to adhere to food-safety best practices during the seafood production chain can trigger microorganism contamination and may result in foodborne diseases [4]. Food-borne infections are a major cause of illness and death worldwide [5,6]. Ani-mal-based foods are widespread all over the world and often considered the key cause of the increase in food-borne infections. Adak et al. [7] find that eating shellfish (a lux-ury food with relatively low consumption levels) is associated with a very high disease risk. Although the number of cases attributed to shellfish are in the same ranges or lev-els as beef or eggs, the level of risk is much higher [8].”

Page 13 LL 490-492: “Little is known about seafood workers' knowledge, beliefs, and habits regarding sea-food handling and safety. Therefore, several studies that assess seafood handling prac-tices by seafood workers are needed, to ensure or improve seafood safety.”

References added:

  1. Soares, K.M.P.; & Gonçalves, A.A. Seafood quality and safety. Instituto Adolfo Lutz 2012, 71(1), 1–10.
  2. Pérez, A.C.A.; Avdalov, N.; Neiva, C.R.P.; Neto, M.J.L.; Lopes, R.G.; Tomita, R.Y. Sanitary-hygienic procedures for industry and seafood inspectors: Recommendations, 2007.
  3. Huss, H.H.; Ababouch, L.; & Gram, L. Assessment andmanagement of seafood safety and quality. Food and Agriculture Orga-nization Fisheries Technical Paper 2003, 444 (230 pp.).
  4. Ifft, J.; Roland-Holst, D.; & Zilberman, D. Consumer valuation of safety-labeled free-range chicken: Results of a field experiment in Hanoi. Agricultural Economics 2012, 43 (6), 607–620.
  5. De Groote, H.; Narrod, C.; Kimenju, S. C.; Bett, C.; Scott, R. P.; Tiongco, M. M., et al. Measuring rural consumers’ willingness to pay for quality labels using experimental auctions: The case of aflatoxin-free maize in Kenya. Agricultural Economics 2016, 47(1), 33–45.
  6. Adak, G. K.; Meakins, S. M.; Yip, H.; Lopman, B. A.; & O’Brien, S. J. Disease risks from foods, England and Wales, 1996–2000. Emerging Infectious Diseases 2005, 11(3), 365.
  7. Gillespie, I. A.; Adak, G. K.; O’Brien, S. J.; Brett, M. M.; & Bolton, F. J. General outbreaks of infectious intestinal disease associated with fish and shellfish, England and Wales, 1992–1999. Communicable Disease and Public Health 2001, 4(2), 117–123.